# Circular RNAs in Epithelial Ovarian Cancer: From Biomarkers to Therapeutic Targets

**DOI:** 10.3390/cancers14225711

**Published:** 2022-11-21

**Authors:** Yumin Qiu, Yan Chen, Oluwatobi Agbede, Esra Eshaghi, Chun Peng

**Affiliations:** 1Department of Biology, York University, Toronto, ON M3J 1P3, Canada; 2Centre for Research on Biomolecular Interactions, York University, Toronto, ON M3J 1P3, Canada

**Keywords:** ovarian cancer, circular RNA, biomarkers, therapeutics

## Abstract

**Simple Summary:**

Epithelial ovarian cancer (EOC) is the deadliest cancer in the female reproductive system. Currently, there are no effective early detection methods for this disease, and the treatments for advanced stages of EOC have a low success rate. Circular RNAs (circRNAs) are noncoding RNAs generated from the “back-splicing” of precursor messenger RNAs. Recent studies have shown that many circRNAs are aberrantly expressed in tumor tissues and/or plasma of EOC patients, and they regulate EOC cell activity and tumor growth. In this review, we provide an overview of EOC and circRNA biology. We then summarize circRNAs that are dysregulated and exert tumor-promoting or tumor-suppressive effects in EOC. We also discuss the possibility of using circRNAs as diagnostic/prognostic markers and therapeutic targets.

**Abstract:**

Epithelial ovarian cancer (EOC) is the most lethal gynecological cancer, and more than 70% of patients are diagnosed at advanced stages. Despite the application of surgery and chemotherapy, the prognosis remains poor due to the high relapse rate. It is urgent to identify novel biomarkers and develop novel therapeutic strategies for EOC. Circular RNAs (circRNAs) are a class of noncoding RNAs generated from the “back-splicing” of precursor mRNA. CircRNAs exert their functions via several mechanisms, including acting as miRNA sponges, interacting with proteins, regulating transcription, and encoding functional proteins. Recent studies have identified many circRNAs that are dysregulated in EOC and may be used as diagnostic and prognostic markers. Increasing evidence has revealed that circRNAs play a critical role in ovarian cancer progression by regulating various cellular processes, including proliferation, apoptosis, metastasis, and chemosensitivity. The circRNA-based therapy may be a novel strategy that is worth exploring in the future. Here, we provide an overview of EOC and circRNA biogenesis and functions. We then discuss the dysregulations of circRNAs in EOC and the possibility of using them as diagnostic/prognostic markers. We also summarize the role of circRNAs in regulating ovarian cancer development and speculate their potential as therapeutic targets.

## 1. Introduction

Ovarian cancer (OC) consists of three major types: epithelial, stromal, and germ cell cancer. Among them, epithelial ovarian cancer (EOC) is the most common one, representing over 90% of cases [1,2]. EOC is the fifth leading cause of cancer-related deaths among women in Canada [3]. Due to the presentation of non-specific symptoms and the lack of effective early diagnosis markers, most EOC patients are diagnosed at advanced stages. Standard chemotherapies for EOC are not effective and patients often develop drug resistance and relapse [1]. Therefore, the development of effective early diagnosis tools and treatment strategies for the late stages of the disease is urgently needed.

Recent studies have revealed that a novel class of RNAs, circular RNAs (circRNAs), plays critical roles in cancer development [4]. CircRNAs are single-stranded RNAs joined covalently at the 5′ and 3′ ends, which are generated from pre-mRNAs via back-splicing [5]. CircRNAs function through multiple mechanisms, most commonly by sponging microRNAs (miRNAs) to affect their target gene expression. Other functions include interacting with proteins to modulate their activities and regulating the transcription of their host genes [6,7,8]. Dysregulation of circRNAs is linked to the development of many diseases, particularly cancers [9]. Aberrant expression of circRNAs occurs in various cancer types, including EOC [10,11]. Emerging evidence suggests that circRNAs are promising diagnostic and therapeutic targets for cancers [12].

In this review, we provide a brief overview of EOC and circRNA biology. We then discuss the potential role of circRNAs as diagnostic and prognostic biomarkers. We also summarize the role of circRNAs in promoting or suppressing EOC development, focusing on circRNAs that have the potential to serve as therapeutic targets for EOC. Finally, we propose some future directions in the field.

## 2. Overview of EOC

### 2.1. Histological Subtypes of EOC

EOC is a highly heterogeneous disease with several major histological subtypes, each one with unique origins and distinct molecular features. The vast majority of EOC tumors can be classified into high-grade serous, low-grade serous, endometroid, mucinous, and clear cell carcinoma [1,13,14,15]. Other rare subtypes include undifferentiated and mixed carcinoma, as well as Brenner tumors [14,16].

High-grade serous carcinoma (HGSC) is the most common subtype of EOC (70–74%) and is responsible for the majority of death from EOC [1]. HGSC contains glandular and papillary cells and also possesses high-grade nuclear atypia, frequent mitotic division, and high chromosomal instability [13]. HGSC mostly develops from non-invasive serous tubal intra-epithelial carcinoma (STIC) originating from a precursor lesion present at the fimbriated ends of the fallopian tubes [17,18]. Prevalent mutation of tumor suppressor genes including *TP53* and *BRCA1/2* commonly occurs in HGSC. Although HGSC tumors are initially sensitive to platinum-based chemotherapy, most patients with advanced-stage cancer will relapse [19].

Low-grade serous carcinoma (LGSC) accounts for less than 5% of EOC and typically develops from serous borderline tumors that could be originated from fallopian tubes [20], endosalpingiosis [15], or ovaries [21]. LGSC tissues are characterized by infrequent mitotic division, an invasive micropapillary pattern, and the presence of round psammoma bodies [22]. Unlike HGSC, mutations of *KRAS*, *HER2*, *BRAF*, and *NRAS* genes are frequently identified in LGSC [15,23]. LGSC patients are typically younger and are resistant to paclitaxel (PTX) and carboplatin drugs, while they are more susceptible to etoposide and doxorubicin-based treatments [24,25].

Endometroid carcinoma (EC) accounts for about 10–15% of EOC [15,26]. It resembles the endometrium in the presence of extensive tubular glands and occasional squamous cells [27]. Although EC likely originates from ovarian endometriosis, more recent studies suggest that it may also develop from the fallopian tubes [28]. Mutations in numerous genes, including *CTNNB1*, *ARID1A*, *TP53*, *PTEN*, *PIK3CA*, *KRAS*, and *PPP2R1A*, have been reported in EC patients [29].

Clear cell carcinoma (CCC) accounts for approximately 6–10% of EOC [15,30]. CCC progression is strongly associated with endometriosis and benign tumors known as clear cell adenofibromas [1]. Essential genes regulating the CCC development pathway are similar to those of EC, including *ARID1A*, *PIK3CA*, *PTEN*, *KRAS*, *PPP2R1A*, *HNF1B*, and *BRCA1/2*. Genetic alterations, including mutation, copy number alteration, and post-translational modification of these genes, contribute to an increased incidence of CCC [1,31]. This subtype is characterized by clear cells, stromal hyalinization, and papillary, tubulocystic and solid cell patterns [31]. CCC has low responsiveness to platinum-based chemotherapy [32].

Mucinous carcinoma (MC) accounts for 3–4% of EOC with a favorable prognosis [15]. Genes with mutation or amplification identified in MC include *KRAS*, *TP53*, *ERBB2*, *BRAF*, *HER2*, *APC*, and *CTNNB1* [1,33]. MC presents as a stratified tissue with atypical nuclei-containing mucins [33]. The growth of MC stems from endometriosis followed by slow and progressive advancement of normal epithelia cells into invasive carcinoma, and this can be explained as the primary developmental pattern of MC. While the distinct origin is unknown, it sometimes originates from other sources including germ cells, teratoma, and metaplasia, and develops at the fallopian tubes, ovaries, or cortical inclusion cysts [15,34].

### 2.2. Stages of EOC

EOC progression is divided into four stages (I to IV) based on the location and metastasis of the cancerous cells as outlined by the International Federation of Gynecology and Obstetrics (FIGO) [35]. The survival rate of patients decreases as cancer progresses from stages I to IV [2]. Stage I tumor is limited to the ovary. The 5-year survival rate at this stage is up to 85–90%. Stage II is characterized by the extension of tumor cells to pelvic organs. By this stage, the 5-year survival rate decreases to about 57–70%. Stage III is marked by tumor cells spreading to the pelvic and abdominal peritoneum, as well as retroperitoneal lymph nodes. Due to the extended metastasis at this stage, the survival rate reduces to 39–59%. Stage IV is defined as distant metastasis (e.g., lung, liver, spleen). At this stage, patients have the worst prognosis and the survival rate is less than 17% [35].

### 2.3. Diagnosis and Treatment

Early diagnosis of EOC is key to reducing morbidity and mortality. Symptoms are usually unhelpful in diagnosing this cancer as they are non-specific. Current imaging techniques, such as transvaginal ultrasound, have limited sensitivity to detect small, early-stage lesions [1]. Two widely used FDA-approved biomarkers, the human carbohydrate antigen 125 (CA125) and the human epididymis protein 4 (HE4), are primarily applied to assist in monitoring cancer progression or recurrence, rather than early diagnosis [36]. CA125 is elevated in only 50% of patients with stage I disease [37], and it is also increased in other gynecological or non-gynecological malignancies [1,38] and benign diseases (such as endometriosis) [39]. HE4 is more specific than CA125, especially in the distinction between endometriosis and the EC subtype [40]; however, the relatively low sensitivity in early diagnosis limits its application [1]. Differential expression patterns of these biomarkers between different cancer subtypes also make it difficult to rely on them as a primary diagnostic tool. Moreover, hormonal contraceptive use, smoking and age all contribute to variations in HE4 levels [39,41,42]. Whether menopausal status could affect the performance of HE4 is still controversial [43].

Currently, surgical resection of the affected region is used as the default treatment. This is usually followed by adjuvant chemotherapy to prevent the recurrence of cancer. In cases where surgical resection is impossible or difficult, chemotherapy could be used before and after surgery to shrink the tumor size [44]. The major concern of this treatment regimen is the toxicity of chemotherapeutic drugs to the cells, as intense cycles of chemotherapy (between 3–6 complete cycles, depending on the cancer stage) are needed to manage EOC. In addition, many patients develop chemoresistance and relapse [45].

Several targeted therapies designed to block abnormal cellular pathways associated with cancerous cell growth in the patient have been under development for EOC. Some of these treatments include small molecule inhibitors, immunotherapy, and antiangiogenic agents. Promising pathways currently exploited in targeted therapy include the vascular endothelial growth factor (VEGF), poly ADP-ribose polymerase (PARP), folate and folate receptor α (FRA), human epithelial growth factor receptor (HER1) and chimeric antigen receptor (CAR) T cell therapy, amidst others [46,47].

PARP inhibitor-based treatment is more widespread and three major inhibitors (olaparib, rucaparib, and niraparib) have been approved by FDA and are currently used to treat patients with EOC [48]. Bevacizumab, a monoclonal antibody targeting VEGF, has also been approved as first-line therapy for EOC patients while multiple VEGF inhibitors are in clinical trials [49]. Combinations of bevacizumab and olaparib targeting both VEGF and PARP have shown efficacy as first-line or second-line maintenance regimens [49]. However, HER1 and CAR T cell-based therapies are still in phase 1 clinical trials [47,50,51].

## 3. Overview of Circular RNA

### 3.1. Biogenesis of circRNAs

The majority of circRNAs are generated from protein-coding genes. While linear mRNAs are processed through canonical eukaryotic pre-mRNA splicing, circRNAs are formed through the back-splicing process, in which a downstream donor splice site is covalently joined to an upstream acceptor splice site in a reverse manner [52]. Most circRNAs are produced from pre-mRNAs and are classified into three groups: exonic circRNAs (ecircRNAs), exon-intron circRNAs (EIciRNAs), and circular intronic RNAs (ciRNAs). EcircRNAs, which consist of one or more exons, are the most common type accounting for 80% of circRNAs [53] and are mostly detected in the cytoplasm [54]. EIciRNAs are derived from exons and introns whereas ciRNAs contain only introns, both of which are in the nucleus [55].

The biogenesis of circRNAs is controlled by trans-regulatory factors and cis-regulatory elements. Although the precise mechanisms are not entirely clear, three models have been proposed to explain how circRNAs are generated (Figure 1). In the lariat-driven circularization model (also known as the exon skipping model), a lariat containing both exons and introns is formed when the upstream acceptor and downstream splice donor get close during the pre-mRNA splicing process. Then introns are spliced, and exons join each other via a covalent bond to form the ecircRNA [53,54]. In some cases, the introns are not removed, which results in the formation of EIciRNAs [56]. When a 7 nt GU-rich sequence is present at the 5′ splicing site and an 11 nt C-rich motif at the 3′ site, a lariat can be formed that only contains introns and eventually leads to the formation of ciRNAs [57]. In the intron-paring model, when the intronic sequences upstream of the splice acceptor and downstream of the splice donor have inverted complementarity, such as the ALU sequence, they join via complementary base pairing and are then excised to form ecircRNAs or EIciRNAs. For the circRNAs to form through this model, each acceptor and donor should have at least 100 nt of ALU complementary sequence [58]. Finally, in the RBP-mediated circularization model, several RNA-binding proteins (RBPs), such as quaking [59] and muscleblind [60], form a bridge to bring the flanking introns of upstream and downstream splice sites into proximity, thereby promoting back-splicing [59,60]. Trans-acting proteins such as spliceosome factors, cleavage factors, and RNA helicases also play a role in regulating circRNA biogenesis [61].

### 3.2. Mechanisms of circRNA Functions

In general, circRNAs exert their biological functions through several mechanisms. The most reported role of circRNAs is to act as a microRNA (miRNA) sponge. miRNAs are small, single-stranded RNAs that primarily inhibit gene expression by binding to the 3′ untranslated region (UTR) of target mRNAs to induce their degradation and inhibit their translation [62]. Numerous studies have reported that circRNAs can bind to miRNAs and this sponging process reduces miRNA availability and thereby releases the inhibitory effect on their target mRNAs, resulting in the upregulation of target genes [6,52]. In addition, many studies have demonstrated that circRNAs interact with proteins, including RBPs and transcription factors [8]. CircRNAs can act as either protein decoys/sponges or scaffolds to affect the functions of the proteins. As protein sponges, they bind to proteins and reduce their interaction with target genes, mRNAs, or interacting proteins. As protein scaffolds, circRNAs can either recruit proteins to specific subcellular locations or facilitate their interaction with other proteins to influence biological processes [53]. The circRNA-protein interaction occurs through specific nucleotide sequences and unique tertiary structures of circRNAs and this is important not only for circRNA functions but also for their biogenesis and degradation [58]. Furthermore, some circRNAs that contain an internal ribosome entry site (IRES), an open reading frame (ORF), and N6-methyladenosine modification can be translated into functional proteins or peptides [53,63], which are mostly found in cancer cells [53]. Finally, EIciRNAs and ciRNAs are mainly located in the nucleus [53]. EIciRNAs interact with U1 snRNP and RNA polymerase II (Pol II) to induce the transcription of their parental genes [53,55]. Similarly, ciRNAs enhance their host gene transcription by associating with Pol II transcriptional machinery [64]. Notably, some ecircRNAs have also been found in the nucleus and regulate their host gene transcription [65].

## 4. CircRNAs as Potential Biomarkers in EOC

CircRNAs are detectable in tissues and body fluids, such as blood, urine, and saliva [66]. Due to their unique circular structures without free 5′ and 3′ ends, circRNAs are resistant to RNA exonucleases and therefore, have a long half-life [67,68]. Recently, accumulating studies have shown the dysregulations of circRNAs in OC and their clinical relevance. The circRNAs that have been reported to be dysregulated in OC and suggested to have diagnostic and prognostic potentials are listed in Table 1. Most of these studies used host gene symbols for circRNAs. In some cases, several circRNAs can be generated from the same precursor mRNA; therefore, we also include the numeric circBase ID in Table 1. Many of these studies did not specify the types and subtypes of OC. Given that the majority of OC cases fall into the EOC group, we include them in the list. However, we limit our discussion to those studies that have good sample sizes, clearly described tumor samples, and have comprehensively analyzed the association between the circRNA and clinicopathologic features, patient survival, and/or chemosensitivity.

### 4.1. CircRNAs as Potential Prognostic Biomarkers

Several circRNAs have been reported to be upregulated in EOC tissues and are associated with poor prognosis of patients. For instance, circABCB10 [88] and circHIPK3 [73] were significantly elevated in EOC tissues compared with the adjacent non-cancerous tissues. Patients with a high level of circABCB10 showed advanced FIGO stage, bigger tumor size, poorer differentiation, and worse overall survival (OS). Similarly, high circHIPK3 levels were associated with advanced stage, lymph node metastasis, and poor survival rate. The Cox regression analysis indicated that circHIPK3 could serve as an independent prognostic factor for worse outcomes of EOC patients. Recently, Li et al. [97] reported that the upregulation of circITGB6 was correlated with low chemosensitivity and poor prognosis in EOC patients. CircITGB6 expression was markedly elevated in tumor tissues obtained from patients with cisplatin (DDP) resistance, as compared with that from chemosensitive patients. In addition, circITGB6 levels were positively associated with advanced FIGO stage, ascites with tumor cells, tumor recurrence, and drug resistance in EOC patients, with most of them having serous carcinomas. Patients with high tumor circITGB6 levels had a higher relapse rate and lower OS. These findings suggest that high levels of circABCB10, circHIPK3, and circITGB6 in EOC tissues may be potential indicators of poor patient outcomes and/or responsiveness to DDP.

Conversely, the downregulation of some circRNAs in EOC tissues may predict a poor prognosis of EOC patients. Ning et al. [106] investigated circRNA expression profiles in EOC specimens compared with normal ovarian tissues through circRNA-sequencing followed by validation using real-time quantitative reverse transcription PCR (qRT-PCR). Five downregulated circRNAs (circBNC2, circEXOC6B, circFAM13B, circN4BP2L2, and circRHOBTB3) and one upregulated circRNA (circCELSR1) were identified in the EOC specimens. Among them, downregulation of circEXOC6B and circN4BP2L2 were significantly associated with advanced FIGO stage and/or lymph node metastasis, as well as poor OS and progress-free survival (PFS), respectively. Similarly, circITCH has also been reported to be downregulated in EOC and other tumors [118,119,120]. Luo et al. [102] confirmed the downregulation of circITCH in EOC tissues compared with paired adjacent samples. The lower circITCH levels were correlated with larger tumor size, advanced FIGO stage and shorter OS. Moreover, circ_0078607 levels were reported to be decreased in HGSC tissues and were negatively correlated with the FIGO stage and serum CA125 levels. The subsequent Cox regression analysis showed that reduced circ_0078607 level was a risk factor for poor PFS and OS [109]. These findings suggest that high levels of circEXOC6B, circN4BP2L2, circITCH, and circ_0078607 are good prognostic markers for EOC patients. However, more studies with a larger group of samples are needed to confirm this. In addition, the correlation between the expression of circRNAs and specific subtypes of EOC remains to be elucidated.

While the circRNAs discussed above raise the possibility of using them as biomarkers for patient prognosis, these studies used adjacent non-cancerous tissues or normal ovarian tissues as controls for the evaluation of circRNA expression levels in EOC. This is not ideal as the majority of EOC tumors originate from epithelial cells outside of the ovary whereas the ovarian tissue contains follicles and stromal tissues and very few epithelial cells [121]. Since circRNAs are expressed in a cell/tissue-dependent manner, the differential expression of circRNAs between EOC and normal ovarian may reflect the different expression levels in different cell types, rather than cancer vs. non-cancerous cells. Therefore, a comparison of circRNAs in serum/plasma between EOC patients and healthy subjects will likely identify more reliable markers. Additionally, liquid biopsy can potentially be repeated as often as necessary to track disease progression and relapse in a real-time manner with minimal harm to patients [122]. Recently, extracellular vesicle (EV)-derived circRNAs in serum have been reported to have a prognostic role in EOC. Luo et al. [99] revealed the potential of circulating exosomal circFOXP1 as a predictor of survival and DDP resistance in EOC. In their study, circFOXP1 was reported to be enriched in exosomes of serum samples and it was strongly upregulated in patients with EOC. Compared with the DDP-sensitive cohort, the expression of exosomal circFOXP1 in DDP-resistant EOC patients was significantly elevated. In addition, the level of exosomal circFOXP1 was positively correlated with FIGO stage, primary tumor size, as well as lymph node and distant metastasis. EOC patients with high circFOXP1 levels presented shorter OS and disease-free survival (DFS), and exosomal circFOXP1 was an independent factor affecting the survival of EOC patients [99]. This study shed light on the application of exosomal circRNAs as prognostic biomarkers in EOC. Similarly, another EV-derived circRNA, circC20orf11, has been identified as a potential prognostic indicator in EOC. Yin et al. [96] demonstrated that increased serum EV-circC20orf11 expression was associated with advanced FIGO stage, high histological grade, and severe lymph node metastasis in EOC patients. EV-circC20orf11 expression was substantially upregulated in the serum of DDP-resistant patients, as compared with that of the DDP-sensitive cohort. The survival rate of patients with enhanced EV-circC20orf11 was significantly decreased. EV-circC20orf11 may act as an indicator of chemotherapy effectiveness and outcome of patients, providing a valuable clue to support the application of EV-derived cargo circRNA as a clinical marker in EOC.

### 4.2. Circulating circRNAs as Potential Diagnostic Biomarkers

To date, no reliable biomarker or panel test exists for EOC diagnosis [1]. Most circRNA biomarker studies in EOC have focused on their prognostic value, while the diagnostic role of circRNAs is rarely studied. Recently, some circRNAs detected in tissue biopsies or blood samples have shown potential in EOC diagnosis. Liquid biopsy is more suitable as a screening tool for early detection due to its advantages of non-invasive, rapid detection and repeatability [122]. Therefore, here we focus only on the diagnostic value of serum/plasma-based circRNAs.

Luo et al. [99] reported that exosomal circFOXP1 was drastically upregulated in the serum of EOC patients and had good diagnostic efficacy in distinguishing EOC patients from healthy volunteers, with the area under curve (AUC) for the receiver operator characteristic (ROC) of 0.914. However, it is unclear whether there is a difference in circFOXP1 levels between healthy subjects and patients with early stages of EOC. Serum circSETDB1 was significantly elevated in patients with serous ovarian tumors, especially in patients with platinum-taxane-combined chemoresistance [98]. Serum circSETDB1 could not only distinguish serous ovarian cancer patients from healthy volunteers with an AUC of 0.8031 (sensitivity 78.33%, specificity 73.33%) but also effectively separate patients with primary chemoresistance from those with primary chemosensitivity (AUC: 0.8107, sensitivity 77.78%, specificity 76.74%) [98], suggesting its potential as a biomarker for diagnosis and chemosensitivity prediction. Since EOC could be easily misdiagnosed as some benign diseases, including endometriosis and pelvic inflammatory mass, it is crucial to consider the efficacy of circRNAs in separating patients with EOC (especially early-stage) from those with benign diseases. However, none of the above studies included patients with benign disorders in the control group, thus the role of these circRNAs may require further verification in optimized control cohorts with potential confounding conditions.

Several recent studies have compared the diagnostic efficacy of plasma circRNAs with CA125 and HE4 in patients. Ge et al. [117] identified two circRNAs, circ_0003972 and circ_0007288, to be considerably downregulated in the plasma of EOC patients. The diagnostic performance of these two circRNAs, either alone or combined (circCOMBO) in distinguishing EOC from benign controls was evaluated. Circ_0007288 (AUC: 0.79), circ_0003972 (AUC: 0.724) or circCOMBO (AUC: 0.781) did not show better diagnostic efficacy than CA125 (AUC: 0.824); however, the combination of CA125 and circCOMBO was more effective (AUC: 0.923) [117], suggesting that the two circRNAs may act as an adjunct to CA125. Excitingly, it was reported recently that plasma circBNC2 was downregulated in EOC patients compared with the benign or healthy cohorts, which might be superior to CA125 and HE4 for EOC detection, especially for early diagnosis [36]. In this study, CA125 showed a poor performance in distinguishing whole-stage (AUC: 0.373) or early-stage (stage I + II) EOC (AUC: 0.204) from the benign cohort, while circBNC2 remained powerful even in the diagnosis of early-stage EOC (AUC: 0.864, sensitivity 92.0%, specificity 80.7%) [36], suggesting that plasma circBNC2 might be a promising novel diagnostic biomarker. Notably, plasma circBNC2 showed the highest sensitivity (95.2%) compared to HE4 (80.7%) and CA125 (24.1%) in the discrimination between EOC and healthy cohorts, and its advantage over CA125 (AUC: 0.204) was the most outstanding in separating early-stage EOC from the benign group [36]. Similarly, Ning et al. [43] reported that plasma circN4BP2L2 could significantly discriminate between EOC and benign or normal cohorts. More importantly, plasma circN4BP2L2 also efficiently separated early-stage EOC from the benign or normal cohort, and the combination of circN4BP2L2 with CA125 and HE4 achieved the best, with an AUC of 0.91 (EOC vs. benign), 89% sensitivity and 87% specificity [43]. However, the study didn’t evaluate the efficiency of combined CA125 and HE4 for early detection, and whether the introduction of circN4BP2L2 could bring significant optimization to the combined CA125 and HE4 was unknown. Notably, circN4BP2L2 performed equally well for EOC diagnosis in both pre- and post-menopausal women, whereas HE4 showed limitations in pre-menopausal cases. Therefore, circN4BP2L2 may serve as a complementary or superior diagnostic biomarker over CA125 and HE4 in EOC.

## 5. circRNAs as Potential Therapeutic Targets

To date, many studies have reported the role of circRNAs in regulating various aspects of EOC development, such as proliferation, metastasis, apoptosis, autophagy, and chemosensitivity. Table 2 lists the circRNAs that are shown to exert either tumor-promoting or tumor-suppressive effects on EOC and the mechanisms underlying their actions. We also include the experimental models in these studies, such as the cell lines used. Since comprehensive reviews on circRNA functions have been published recently [10,123,124], we focus on circRNAs that are strongly dysregulated in EOC and have been studied using multiple model systems, especially animal models, and discuss their therapeutic potential.

### 5.1. circRNAs That Exert Tumor-Promoting Effects in EOC

Most circRNAs studied in EOC to date have been reported to exert tumor-promoting effects by inducing proliferation, metastasis, and/or reducing apoptosis via sponging various miRNAs. For instance, circ_0000144 was shown to act as a competing endogenous RNA (ceRNA) for miR-610 to induce ELK3 expression, and subsequently promote EOC cell proliferation, migration and invasion, as well as tumor formation in nude mice. This circRNA was also elevated in tumor samples and serum of EOC patients and was negatively associated with patient survival [93]. Similarly, circ_0004712 was increased in EOC tissues compared with adjacent non-cancerous samples and was associated with high grade of tumor and lymph node metastasis, as well as poor outcomes of patients. Knockdown of circ_0004712 impaired proliferation, migration, invasion, viability, and tumor formation. Mechanistically, circ_0004712 acts as a tumor promoter by binding to miR-331-3p to upregulate its target gene, *FZD4* [94]. Although these findings suggest that circ_0000144 and circ_0004712 have tumor-promoting effects, more studies with larger sample sizes and histological subtype-matched cell lines are needed to further evaluate their therapeutic potential.

Several circRNAs have recently been identified to promote EOC growth and metastasis by interacting with proteins or by potentially translating into a functional protein. CircRHOC was markedly upregulated in tumors, and it induced EOC cell viability, migration, and invasion in vitro. Silencing of circRHOC significantly reduced the size and number of tumor nodes in the mesentery of mice intraperitoneally injected with EOC cells, suggesting the promoting effect of circRHOC on tumor dissemination. These effects are likely mediated through both indirect and direct regulation of VEGF. First, circRHOC acts as a miR-302e sponge to upregulate *VEGFA*, which is a target gene of miR-302e. Second, circRHOC may directly bind and modulate VEGFA expression in EOC [78]. Chen et al. [76] discovered that circNOLC1, which was upregulated in EOC tumors when compared with benign and normal ovaries, promoted cell proliferation, migration, and invasion, but inhibited apoptosis in EOC in vitro. Using a subcutaneous xenograft mouse model, they found that EOC cells transfected with short hairpin RNA (shRNA) targeting circNOLC1 formed smaller tumors than the control cells, suggesting that knockdown of this circRNA inhibits EOC development. Mechanistically, circNOLC1 exerts these effects likely by interacting with ESRP1 protein to upregulate downstream CDK1 and RhoA. Additionally, Li et al. [150] showed that circ_0001756 exhibited various tumor-promoting effects on EOC via binding with IGF2BP2 protein. Circ_0001756 could bind and positively regulate IGF2BP2 and RAB5A to activate the downstream EGFR/MAPK signaling pathway, thereby inducing cell proliferation, invasion, and epithelial-mesenchymal transition (EMT) in vitro and tumor growth in vivo. Interestingly, circCRIM1 was discovered to enhance EOC cell viability and metastasis not only by sponging miR-145-5p and miR-383-5p but may also by translating into a protein. CircCRIM1 harbors an ORF with 188 amino acids spanning the splice junction. Overexpression of this protein induced cell viability, migration and invasion, and inhibited apoptosis in EOC, suggesting a tumor-promoting activity of the circCRIM1-encoded protein [171]. However, whether circCRIM1 is endogenously translated into the protein remains to be confirmed.

Tumor cells in ascites are known to play a major role in EOC metastasis [190,191]. EOC cells disseminate from the primary site to ascites through EMT and transfer to the peritoneal cavity via ascites circulation, where they form metastatic lesions [190,192,193]. A recent study identified upregulated circ_0000497 and circ_0000918 in ascites tumor cells (ASCs) when compared with cells from primary and metastatic tumors in patients with HSGC. Transcriptomic analyses revealed that the ASCs exhibited mesenchymal phenotypes while the tumor cells from primary and metastatic lesions displayed gene expression patterns of epithelial cells. In vitro studies showed that these two circRNAs promoted migration and invasion, downregulated ZO1 and E-cadherin, and upregulated N-cadherin and vimentin in EOC cells [161], suggesting that the circRNAs may contribute to EOC metastasis by modulating EMT. However, the therapeutic potential of these circRNAs remains to be evaluated in vivo.

Like EMT, mesothelial-to-mesenchymal transition (MMT) also contributes to peritoneal metastasis of EOC and exosomes from primary tumor cells are involved in this process [194]. CircWHSC1 [170] and circPUM1 [77] were reported to not only promote EOC cell proliferation, migration and invasion by sponging miRNAs but also act on the peritoneum in an exosome-dependent manner. When circWHSC1- and circPUM1-rich exosomes isolated from EOC cells were injected into mice grafted intraperitoneally with EOC cells, the peritoneal mesothelial cells were converted into carcinoma-associated fibroblasts in mice, thus providing a suitable environment for peritoneal implantation and dissemination of EOC in vivo. These studies suggest that targeting circWHSC1 and circPUM1 may be a potential therapeutic strategy for patients with metastatic EOC.

Angiogenesis increases intratumoral perfusion of nutrients and oxygen and plays an essential role in solid tumor growth and metastasis [1,195]. CircASH2L and circPTK2 have been reported to promote angiogenesis and tumor progression in EOC. Chen et al. [86] showed that circASH2L was remarkably upregulated in tumor tissues, and was associated with advanced FIGO stage, large tumor size, and lymph node metastasis. CircASH2L stimulated VEGFA expression by sponging miR-665, thereby enhancing the invasion and proliferation of EOC cells and angiogenic activity of human umbilical vein endothelial cells (HUVECs) in vitro. Notably, silencing of circASH2L hindered tumor-associated angiogenesis and lymphangiogenesis, and repressed tumor growth in mice. Additionally, Wu et al. [155] reported that the expression of circPTK2 was significantly higher in tumors than in normal ovarian tissues. CircPTK2 promoted migration, invasion and EMT of EOC cells, and also induced angiogenesis capability of HUVECs in vitro. Overexpression of circPTK2 accelerated EOC tumorigenesis in a mouse xenograft model. These effects were mediated through the miR-639/FOXC1 axis. Therefore, targeting circASH2L and circPTK2 using small interfering RNA (siRNA) or shRNA may serve as a potential anti-angiogenic strategy in EOC.

Autophagy is another critical process in EOC development. Autophagy could play a tumor-suppressive role by reducing the damaged organelles (e.g., mitochondria) that lead to genomic instability and inducing autophagic cell death [196,197,198]. On the other hand, autophagy also increases the tolerance to nutrient deprivation and maintains the survival of tumor cells [199,200]. Recently, circRNF144B was reported to promote EOC progression by inhibiting tumor-suppressing autophagy. CircRNF144B was upregulated in tumor specimens in comparison to that in adjacent non-tumor tissues. Knockdown of circRNF144B increased autophagy and reduced tumor cell proliferation, migration and invasion in vitro. When circRNF144B knockdown cells were subcutaneously inoculated into nude mice, they formed small tumors than the control cells. Through tail vein injection, it was also found that the circRNF144B-silenced cells had lower metastatic potential. Further experiments showed that circRNF144B could sponge miR-342-3p to induce FBXL11 expression, leading to the ubiquitination and degradation of Beclin-1 and subsequent suppression of autophagy and EOC progression [92]. On the other hand, circMUC16 and circRAB11FIP1 were reported to induce tumor-promoting autophagy in EOC. CircMUC16 expression was higher in tumors and serum of EOC patients and was positively associated with advanced FIGO stage and tumor grade. CircMUC16 induced autophagy flux and LC3-II accumulation in EOC cells, and the circMUC16-mediated autophagy promoted cellular proliferation and invasion [84]. Silencing of circMUC16 suppressed pelvic peritoneal metastasis in EOC tumor-bearing mice. Mechanistically, circMUC16 induced autophagy via a circMUC16/miR-199a-5p/RUNX1 feedback loop [84]. Similarly, circRAB11FIP1 was found to promote autophagy of EOC through interacting with the miR-129 and DSC1. This circRNA was upregulated in EOC tissues and induced autophagy to accelerate cellular proliferation and invasion, while knockdown of which inhibited tumor dissemination in the peritoneal cavity of mice. CircRAB11FIP1 upregulated ATG14 and ATG7 by sponging miR-129 and by binding to DSC1 to facilitate its interaction with ATG101, thereby inducing autophagy of EOC to promote tumor progression [157].

Chemotherapy resistance is one of the main causes of poor survival in EOC patients [201]. Several circRNAs were discovered to contribute to chemoresistance in EOC. Using microarray and subsequent qPCR analyses, researchers found circTNPO3 to be one of the most strongly upregulated circRNAs in PTX-resistant EOC tissues when compared with PTX-sensitive EOC tumors. Silencing of circTNPO3 significantly enhanced the responsiveness of EOC cells to the PTX treatment both in vivo and in vitro. These effects are likely mediated via the miR-1299/NEK2 pathway [89]. In another study, circCELSR1 was one of the most highly upregulated circRNAs in PTX-resistant EOC tumors. CircCELSR1 targeted the miR-1252/FOXR2 axis to enhance the PTX resistance of tumor cells. Knockdown of circCELSR1 increased the cytotoxic activity of PTX in vitro and enhanced PTX-mediated tumor growth inhibition in mice [129]. Additionally, Li et al. [97] illustrated that circITGB6 enhanced the DDP resistance of EOC through a mechanism involving tumor-associated macrophages (TAMs). CircITGB6 was identified to be highly upregulated in tumors and serums of EOC patients resistant to DDP treatment. Overexpression of circITGB6 reduced while silencing of circITGB6 enhanced, the antitumor effect of DDP in vivo. Further examination revealed that circITGB6 induced macrophage polarization toward the pro-tumor M2 phenotype through the direct interaction with IGF2BP2 to stabilize *FGF9* mRNA. The circITGB6-mediated TAM M2 polarization resulted in an immunosuppressive microenvironment and thus led to DDP resistance of EOC. Interestingly, intraperitoneal injection of an antisense oligonucleotide (ASO) targeting circITGB6 into mice bearing EOC tumors significantly inhibited M2 polarization, enhanced DDP sensitivity and increased the OS rate of mice. These studies raise the possibility of targeting these circRNAs to overcome chemoresistance in EOC.

### 5.2. circRNAs That Exert Tumor-Suppressing Effects in EOC

Several circRNAs have been reported to suppress tumor development in EOC. Recently, Wu et al. [114] discovered that circFBXO7 inhibited malignancies of EOC cells by inhibiting Wnt/β-catenin activity. Through rRNA-depleted RNA-seq, circFBXO7 was identified to be highly downregulated in EOC tissues, as compared with normal ovary samples. Functional analysis showed that overexpression of circFBXO7 repressed EOC cell proliferation, migration and invasion, while knockdown of circFBXO7 had the opposite effect. Importantly, circFBXO7 overexpression suppressed both local tumor growth and metastasis in mice that had been inoculated with EOC cells. CircFBXO7 exerts these anti-tumor effects by acting as a ceRNA of miR-96-5p to upregulate its target gene *MTSS1*, which in turn, inhibits the activity of the Wnt/β-catenin pathway [114].

CircRHOBTB3 (circ_0007444) was reported to inhibit EOC progression by serving as a ceRNA to facilitate PTEN expression [110,111]. PTEN is a critical tumor suppressor in a variety of cancers, including EOC [202,203,204,205]. Wu et al. [111] showed that circ_0007444 was markedly decreased in tumor samples, as compared with normal tissues. Patients with low circ_0007444 expression presented with advanced tumor stages and grades, large tumor sizes, and low 5-year survival. Circ_0007444 attenuated proliferation, migration and invasion, and induced apoptosis of EOC cells. Overexpression of circ_0007444 also substantially inhibited tumor growth in nude mice. These effects were mediated via targeting miR-570-3p to upregulate PTEN expression. Similarly, Fu et al. [110] reported that circRHOBTB3 was downregulated in EOC tissues, and its expression was negatively correlated with advanced FIGO stage, lymph node metastasis and poor OS. CircRHOBTB3 could absorb miR-23a-3p to upregulate PTEN, thereby blocking AKT activation. Mediated by the miR-23a-3p/PTEN/AKT axis, circRHOBTB3 suppressed proliferation, G1/S transition, invasion, and induced apoptosis in EOC cells. Overexpression of circRHOBTB3 also greatly inhibited tumor growth in nude mice. These studies suggest that circRHOBTB3 exerts a potent tumor-suppressive effect by upregulating PTEN. *PTEN* loss or mutations are common in EC and CCC [1]. PTEN loss or downregulation is also a frequent event in HGSC that defines a molecular subgroup with a significantly worse prognosis [206]. Therefore, targeting circRHOBTB3 to reconstitute PTEN function might be a potential therapeutic strategy for EOC.

CircCDR1as, also known as ciRS-7, is one of the most well-studied circRNAs. CircCDR1as is mostly regarded as a tumor-promoting circRNA in a variety of cancers, but its role in EOC is still controversial. Three studies reported that circCDR1as enhanced chemosensitivity and reduced cell proliferation, migration, and invasion [70,71,72], while another study showed that circCDR1as promoted EOC progression [69]. Zhao et al. [71] observed that circCDR1as sensitized EOC to DDP by modulating the miR-1270/SCAI signaling pathway. CircCDR1as was remarkably downregulated in tumor tissues and serum exosomes from OC patients with DDP resistance, as compared with that from DDP-sensitive patients. CircCDR1as overexpression enhanced while its knockdown reduced basal and DDP-induced apoptosis in vitro. Silencing of circCDR1as also accelerated tumor growth in vivo and strongly attenuated the inhibitory effect of DDP on tumor growth [71], suggesting its potential involvement in the response of tumor cells to DDP. CircCDR1as could bind to miR-1270 to weaken its suppression on the downstream SCAI, thereby enhancing the DDP chemosensitivity of EOC [71]. Similar effects of circCDR1as on increasing DDP activity in vitro and in vivo were reported in another study, in which miR-1299 and its target gene, *PPP1R12B*, were identified as downstream effectors [70]. CircCDR1as has also been reported to have lower expression in EOC tumors and inhibit cell proliferation, migration, and invasion in vitro [72]. In contrast, Zhang et al., reported that circCDR1as was upregulated in EOC tissues and its knockdown decreased cell proliferation, migration, and invasion in vitro and tumor growth in vivo [69]. The reason for these opposite findings is unknown. One possibility may be due to the different histological subtypes of the samples examined. The controls used in these two studies were ovarian epithelial tissues from non-cancer patients and ovarian tissues adjacent to the tumors, respectively. As these control samples have different cell types, this may have contributed to such discrepancies. Also, techniques used in circRNA research, such as qPCR and si/shRNA-mediated circRNA knockdown, have limitations. Therefore, more studies are required to determine the involvement of circCDR1as in EOC development and its potential as a therapeutic target.

CircPLEKHM3 has been reported to have tumor-suppressive effects on EOC [115,177]. Sun et al. [177] showed that circPLEKHM3 exacerbated the curcumin-mediated tumor suppression by targeting the miR320a/SMG1 axis. CircPLEKHM3 was significantly downregulated in tumors compared with normal ovarian tissues. CircPLEKHM3 enhanced the cytotoxicity of curcumin in EOC cells and strengthened the anti-tumor effect of curcumin in mice. In addition, Zhang et al. [115] reported that circPLEKHM3 inhibited EOC progression through the modulation of a miR-9/BRCA1/DNAJB6/KLF4/AKT1 axis. Using RNA-seq, qRT-PCR, and single-molecule RNA in situ hybridization, the authors found that circPLEKHM3 was strongly downregulated in tumors when compared with that of normal ovarian tissues and was also reduced in peritoneal metastatic ovarian tumors as compared with the corresponding primary lesions. Downregulation of circPLEKHM3 was correlated with short OS and recurrence-free survival in EOC patients. On the other hand, the linear *PLEKHM3* mRNA expression did not differ between tumor tissues and normal samples and was not associated with patient outcomes, suggesting that circPLEKHM3 acts as a tumor suppressor independent of its linear transcript. In vitro and in vivo studies demonstrated that circPLEKHM3 played an anti-tumor role by sponging miR-9 to enhance the tumor-suppressive effect of BRCA1, DNAJB6, and KLF4, and consequently inactivated the AKT signaling in EOC. However, it remains to be investigated whether circPLEKHM3 overexpression could inhibit tumor formation or reduce tumor burden and therefore, serves as a potential therapeutic strategy for EOC.

CircATRNL1 is known to inhibit EOC development by acting as a ceRNA or translating into a protein. Wang et al. [107] reported that circATRNL1 was significantly reduced in tumor tissues, and its low expression was correlated with advanced stage, lymph node metastasis, and poor survival rate of patients. CircATRNL1 exerted anti-tumor functions by targeting miR-378 to activate the SMAD4 pathway. Overexpression of circATRNL1 restrained the proliferation, migration and invasion of EOC cells and inhibited angiogenesis of HUVECs in vitro. In a mouse intraperitoneal xenograft model, overexpression of circATRNL1 significantly reduced the number and weight of tumor nodules in the abdomen, suggesting that circATRNL1 effectively represses tumor growth and metastasis in vivo. Additionally, Lyu et al. [179] suggested that circATRNL1 may act as a tumor suppressor by encoding a 131 amino acid protein in EOC cells. CircATRNL1 was downregulated in tumor samples from stage III-IV HGSC patients, as compared with paired normal ovarian tissues, and it inhibited proliferation, migration, and invasion of EOC cells. CircATRNL1 harbors an ORF and an IRES that may initiate ORF translation. However, whether circATRNL1 is endogenously translated into the protein and the biological activity of this encoded protein remain to be confirmed.

## 6. Concluding Remarks and Future Directions

Increasing evidence has suggested that dysregulation of circRNAs contributes to EOC development. However, research on circRNA in EOC is still at an early stage and much work needs to be done to develop circRNA-based biomarkers and therapeutics. The up- or downregulation of certain circRNAs in EOC tissues or patient serum samples and their association with patient survival, cancer stages and/or grades, and chemosensitivity suggest that circRNAs could be used as diagnostic or prognostic markers of EOC. However, most studies published to date did not specify histologic subtypes of EOC. Therefore, circRNAs dysregulated in each subtype of EOC and their correlations with the genomic characteristics remain to be explored. Given the heterogeneous origins of EOC, a single circRNA biomarker is highly unlikely to be sufficient for all subtypes, and circRNA panels will probably be more robust. In the future, multi-center prospective studies with large sample cohorts with defined histology subtypes and EOC stages will likely facilitate the transformation of circRNA research from the laboratory to the clinic. Also, the potential of using circRNAs as predictive biomarkers of EOC remains to be elucidated. It might also be interesting to establish algorithms based on circRNAs and their combination with other markers, such as HE4 and CA125. The risk of ovarian malignancy algorithm (ROMA), which combines HE4 and CA125 measurements with menopausal status, has been approved by the FDA to determine the risk of malignancy in women with pelvic mass [1,207,208,209]. It will be interesting to determine if the introduction of circRNAs into the ROMA algorithm may lead to the development of effective screening or early detection tools for EOC.

Most studies reported so far used qRT-PCR to quantify circRNAs in cells and tissues. However, since this technique relies on reverse transcription and PCR amplification with divergent primers, there could be experimental bias and artifacts [210,211]. The nanoString nCounter, a novel digital counting technology, is already in clinical use for breast cancer prognostication based on the prediction analysis of microarray 50 (PAM50) mRNA test [212,213]. With color-coded probes spanning the specific back-spliced junctions of circRNAs, the nanoString nCounter is also applicable for circRNAs detection [210]. This novel enzyme-free approach was proven to be superior to qRT-PCR in sensitivity, specificity, and quantitative accuracy of circRNA detection [210]. It is reliable even in low-quality RNA samples such as those from severely degraded formalin-fixed paraffin-embedded tissues [210]. In the future, the application of novel technologies in EOC studies will likely accelerate the discovery and implementation of circRNA-based biomarkers in clinical practice.

So far, more than ninety circRNAs have been reported to regulate EOC cell proliferation, apoptosis, migration, invasion, chemosensitivity, and/or tumor growth. The research strategies employed in these studies generally include the measurement of circRNA levels and their clinicopathological relevance in patient samples, functional assessment using knockdown and overexpression approaches, and xenograft mouse models to evaluate the in vivo functions of circRNAs. Determination of the interacting molecules (e.g., miRNAs and proteins) and their downstream target genes is also investigated. For in vitro models of these studies, SKOV3 and A2780 are the most frequently used cell lines. SKOV3 is also the most used cell line for in vivo models (Table 2). These cell lines represent the EC subtype of EOC [1]. Given that HGSC is the most common subtype, more cell lines derived from HGSC should be used. Some of the cell lines used are problematic. For example, HO-8910 cells have been reported to be contaminated with cervical adenocarcinoma HeLa cells [214,215]. The OV-1063 cell line might be partially misidentified (www.atcc.org accessed on 27 October 2022), and the SW626 cell line may be established from an ovarian metastasis of a colonic carcinoma [216]. Therefore, it is important to not only use multiple cell models representing different subtypes but also select appropriate cell lines for EOC studies.

Functional studies of circRNAs in EOC have been commonly carried out using overexpression and knockdown approaches. Overexpression is typically achieved by transfecting a circRNA-expressing vector into EOC cells. However, it has been reported that circRNA-expressing plasmids could also co-produce linear RNAs [217]. Therefore, to make sure that the phenotypes observed are indeed due to the circRNA overexpression, it is recommended to measure both circ- and linear RNAs and/or include a construct lacking one or two intronic repeats as a control vector, which cannot be spliced into the circRNA [217]. In loss-of-function experiments, siRNAs or shRNAs targeting the back-splice junction are used. These si/shRNAs may also result in reduced levels of the cognate linear mRNAs [217,218]. Recently, CRISPR/Cas13 system, a more specific knockdown technique, has been applied to silence circRNAs [218,219] and this could be used in future EOC studies to reduce off-target effects in loss-of-function assays. Another circRNA depletion strategy, which uses ASO to target the back-splicing junction of circRNA, has also shown promising tumor-suppressing effects in other cancers [220,221]. This strategy has been reported recently to inhibit circITGB6 in EOC [97]. However, more studies are needed to further investigate the therapeutic potential of ASO-based strategy in EOC.

To study the in vivo effects of circRNAs in EOC development, researchers inoculated mice with tumor cells with stable overexpression or knockdown of a circRNA. Other in vivo delivery strategies with clinical potential, including nanoparticle delivery systems, remain to be explored in EOC. Nanoparticle delivery systems have shown good delivery efficiency for circRNAs in other cancers. Du et al. [222] reported that injection of a siRNA targeting circDnmt1 conjugated with gold nanoparticles inhibited breast cancer progression and extended the lifespan of mice. Lu et al. [223] have shown that circEHMT1 plasmid-loaded nanoparticles inhibited lung metastasis of breast cancer in vivo. Therefore, nanoparticle delivery of circRNA-based agents will be useful to investigate the therapeutic potential of circRNAs in EOC in the future.

In summary, although some circRNAs have been proposed to play important roles in EOC and have diagnostic and therapeutic potentials, there are still lots of challenges in this field. In the future, more rigorous and comprehensive studies are needed to identify biomarkers for EOC and to clarify the functions and mechanisms of circRNAs. Development of in vivo delivery strategies of circRNAs is also required to advance their clinical potential in EOC. Although there is more work to be done, the development and application of circRNA-based prediction, diagnosis, and prognostic assessment and therapeutics could lead to early detection of EOC and help guide clinicians to make optimal personalized treatment decisions, monitor disease progression and recurrence, and thus greatly improve the survival and life quality of EOC patients.

## Figures and Tables

**Figure 1 cancers-14-05711-f001:**
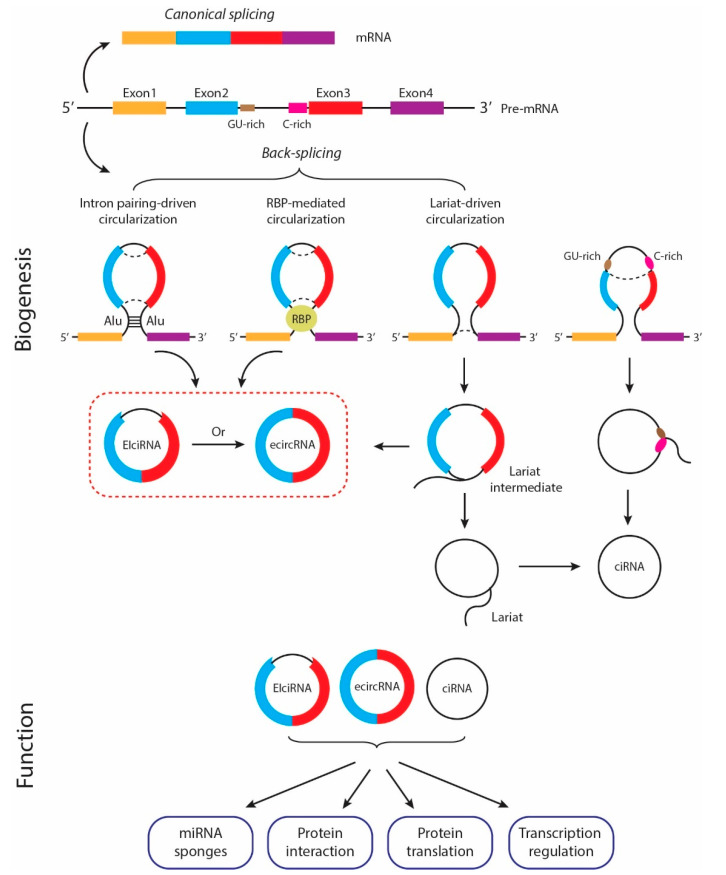
The biogenesis and functions of circRNAs. CircRNAs can be generated through multiple mechanisms. In the lariat-driven circularization model, a lariat is formed during pre-mRNA splicing to generate either ecircRNA, EIciRNA, or ciRNA. In the intron pairing-driven circularization model, the inverted complementary sequences between the upstream acceptor and downstream donor join via complementary base pairing, leading to the formation of EIciRNA or ecircRNA. In the RBP-driven model, some RBPs bring the acceptor and donor to proximity and promote circularization, resulting in the formation of EIciRNA or ecircRNA. CircRNAs exert their biological functions through several mechanisms. They can sponge miRNAs to inhibit their functions or interact with proteins directly to regulate their activity. Some circRNAs also translate functional proteins/peptides, or function in the nucleus by regulating their parental gene transcription.

**Table 1 cancers-14-05711-t001:** Dysregulated circRNAs in OC and their potential as biomarkers.

circRNA (circBase ID)	Level in OC	Detection Methods, Cohort Size/Subtype	Sample Source	Clinicopathologic Features	Prognosis	Diagnosis	Refs
circCDR1as (ciRS-7, circ_0001946)	Up	qRT-PCR: OC vs. adjacent normal tissues (*n* = 40)	Tissues	TNM stage, metastasis	OS	-	[69]
	Down	qRT-PCR: OC vs. adjacent normal tissues (*n* = 6)	Tissues	-	-	-	[70]
	Down	Microarray: DDP-resistant vs. DDP-sensitive SOC (*n* = 5). qRT-PCR: DDP-resistant SOC (tissues and serum exosome, *n* = 36), DDP-sensitive SOC (tissues and serum exosome, *n* = 30), adjacent normal tissues (*n* = 66).	Tissues, serum	DDP resistance	-	-	[71]
	Down	qRT-PCR: OC (*n* = 65), normal ovarian epithelial tissues (*n* = 37)	Tissues	-	-	-	[72]
circHIPK3 *	Up	qRT-PCR: EOC vs. adjacent non-cancerous tissues (*n* = 69)	Tissues	FIGO stage, metastasis	OS, DFS	-	[73]
circCEACAM5 (circ_0051240)	Up	Microarray: OC vs. adjacent normal tissues (*n* = 10) qRT-PCR: OC vs. adjacent non-cancerous tissues (*n* = 33)	Tissues	FIGO stage, metastasis	OS	-	[74]
circCSPP1 (circ_0001806)	Up	qRT-PCR: EOC (*n* = 117), borderline tumors (*n* = 12), normal ovarian tissues (*n* = 15), benign ovarian tumors (*n* = 12)	Tissues	FIGO stage, differentiation	-	-	[75]
circNOLC1 (circ_0000257)	Up	qRT-PCR: EOC (86 serous, 32 others), normal ovarian tissues (*n* = 15), benign ovarian tumors (*n* = 11), borderline ovarian tumors (*n* = 11)	Tissues	FIGO stage, differentiation, serum CA125 level	-	-	[76]
circPUM1 (circ_0000043)	Up	qRT-PCR: EOC (*n* = 62), normal ovarian tissues (*n* = 13)	Tissues	FIGO stage	-	-	[77]
circRHOC (circ_0013549)	Up	qRT-PCR: OC (100 serous carcinomas, 27 others), normal ovarian tissues (*n* = 24)	Tissues	FIGO stage, differentiation	-	-	[78]
circCELSR1 (circ_0063804)	Up	qRT-PCR: OC vs. adjacent normal tissues (*n* = 108)	Tissues	FIGO stage, tumor grade, tumor size	OS	-	[79]
circSPECC1 (circ_0000745)	Up	qRT-PCR: OC (35 EOC, 9 germ cell tumors, 6 others), para-cancerous tissues (*n* = 50)	Tissues	FIGO stage, tumor grade	-	-	[80]
circHIPK2 (circ_0001756)	Up	qRT-PCR: OC (26 DDP-sensitive, 20 DDP-resistant), adjacent normal tissues (*n* = 46)	Tissues	FIGO stage, tumor size, metastasis, DDP resistance	-	-	[81]
circMFN2 (circ_0009910)	Up	qRT-PCR: OC (*n* = 50), normal ovarian tissues (*n* = 50)	Tissues	FIGO stage, metastasis	OS	-	[82]
circMYLK *	Up	qRT-PCR: OC vs. adjacent normal tissues (*n* = 46)	Tissues	TNM stage	OS	-	[83]
circMUC16 (circ_0049116)	Up	RNA-seq: EOC (*n* = 3), healthy ovarian tissues (*n* = 4). qRT-PCR: EOC (*n* = 70), normal ovarian tissues (*n* = 30); serum of EOC patients (*n* = 70), serum of healthy subjects (*n* = 30)	Tissues, serum	FIGO stage, tumor grade	-	-	[84]
circFAM53B (circ_0000267)	Up	qRT-PCR: EOC (36 serous, 18 mucinous), paired non-cancerous tissues (*n* = 54)	Tissues	FIGO stage, tumor size, metastasis	OS	-	[85]
circASH2L (circ_0006302)	Up	qRT-PCR: OC vs. adjacent normal tissues (*n* = 50)	Tissues	FIGO stage, tumor size, metastasis	-	-	[86]
circACP6 (circ_0013958)	Up	qRT-PCR: serous ovarian carcinomas (SOC, *n* = 45), normal ovarian tissues (*n* = 45)	Tissues	FIGO stage, metastasis	-	AUC = 0.912	[87]
circABCB10 *	Up	qRT-PCR: EOC (67 serous, 36 others), adjacent tissues (*n* = 53)	Tissues	FIGO stage, differentiation, tumor size	OS	AUC = 0.766	[88]
circTNPO3 (circ_0001741)	Up	Microarray: PTX-sensitive vs. PTX-resistant OC (*n* = 3) qRT-PCR: OC (20 PTX-sensitive, 28 PTX-resistant), adjacent normal tissues (*n* = 48)	Tissues	FIGO stage, PTX resistance	OS	AUC = 0.910	[89]
circANKRD17 (circ_0007883)	Up	RNA-seq: EOC (*n* = 3), healthy ovarian tissues (*n* = 4) qRT-PCR: EOC (28 serous, 20 mucinous; 28 PTX resistant, 20 PTX sensitive), adjacent normal tissues (*n* = 48)	Tissues	PTX resistance	OS	-	[90]
circEEF2 (circ_0048559)	Up	FISH: EOC (40 serous, 9 mucinous, 8 clear cell), normal ovarian tissues (*n* = 12)	Tissues	FIGO stage, tumor grade	-	-	[91]
circRNF144B (circ_0075797)	Up	RNA-seq: OC tissues with high and low LC3 dots (*n* =3) qRT-PCR: OC vs. adjacent non-cancerous tissues (*n* = 36)	Tissues	Metastasis	OS	-	[92]
circSLAMF6 (circ_0000144)	Up	qRT-PCR: OC tissues vs. adjacent normal tissues (*n* = 60); serum of OC patients vs. serum of healthy (*n* = 60)	Tissues, serum	-	OS, DFS	-	[93]
circPDE7B (circ_0004712)	Up	qRT-PCR: EOC (20 serous, 10 mucinous) vs. adjacent non-cancerous tissues (*n* = 30)	Tissues	Tumor grade, metastasis	OS, PFS	-	[94]
circC20orf11 (circ_0061140)	Up	qRT-PCR: OC vs. adjacent normal tissues (*n* = 55)	Tissues	TNM stage, metastasis, tumor size	-	-	[95]
circC20orf11 *	Up	qRT-PCR: OC (47 serous, 13 others; 26 DDP-sensitive, 34 DDP-resistant)	Serum	FIGO stage, tumor grade, metastasis, DDP resistance	OS	-	[96]
circITGB6 (circ_0056856)	Up	RNA-seq: DDP-sensitive vs. resistant OC tissues (*n* = 5) qRT-PCR: Serum of OC (20 DDP-resistant, 20 DDP-sensitive), serum of normal controls (*n* = 15); OC tissues (20 DDP-resistant, 20 DDP-sensitive). ISH: EOC tissues (95 SOC, 18 MC, 3 EC, 2 CCC, 1 other)	Serum, tissue	FIGO stage, tumor recurrence, DDP resistance	OS, RFS	-	[97]
circSETDB1 (circ_0006352)	Up	qRT-PCR: SOC (18 primary chemoresistance, 42 primary chemosensitive), healthy (*n* = 60).	Serum	FIGO stage, metastasis, chemosensitivity	PFS	SOC vs. healthy: AUC = 0.803 Chemoresistant vs. chemosensitive SOC: AUC = 0.811	[98]
circFOXP1 (circ_0001320)	Up	qRT-PCR: EOC (46 DDP-resistant, 76 DDP-sensitive), healthy (*n* = 82)	Serum	FIGO stage, metastasis, tumor size, DDP resistance	OS, DFS	AUC = 0.914	[99]
circMGAT5 (circ_0001068)	Up	Microarray: OC vs. healthy (*n* = 4). qRT-PCR: (1) training set: OC (*n* = 10), healthy (*n* = 10); (2) larger cohort: OC (*n* = 85), healthy (*n* = 43).	Serum	-	-	AUC = 0.970	[100]
circITCH *	Down	qRT-PCR: OC vs. adjacent normal tissues (*n* = 45)	Tissues	FIGO stage, tumor size	OS	-	[101]
circITCH *	Down	qRT-PCR: EOC (48 serous, 29 others) vs. adjacent normal tissues (*n* = 77)	Tissues	FIGO stage, tumor size	OS	-	[102]
circRNA_100395 (circ_0015278) #	Down	qRT-PCR: OC vs. adjacent normal tissues (*n* = 60)	Tissues	FIGO stage, metastasis	OS	-	[103]
circLARP4 *	Down	qRT-PCR: EOC vs. adjacent normal tissues (*n* = 78)	Tissues	FIGO stage, metastasis.	OS, DFS	-	[104]
circEXOC6B (circ_0009043)	Down	qRT-PCR: OC vs. adjacent normal tissues (*n* = 60)	Tissues	TNM stage, metastasis	OS	-	[105]
circEXOC6B (circ_0009043)	Down	RNA-seq: EOC vs. normal ovarian tissues (*n* = 4) qRT-PCR: EOC vs. normal ovarian tissues (*n* = 54)	Tissues	Metastasis	OS	-	[106]
circATRNL1 (circ_0020093)	Down	qRT-PCR: OC (30 serous, 26 others), adjacent non-cancerous tissues (*n* = 56)	Tissues	FIGO stage, metastasis	OS	-	[107]
circSLC22A3 (circ_0078607)	Down	qRT-PCR: OC vs. adjacent normal tissues (*n* = 43)	Tissues	FIGO stage, metastasis	-	-	[108]
		qRT-PCR: HGSC vs. adjacent non-cancerous tissues (*n* = 49)		FIGO stage, serum CA125 level	OS, PFS		[109]
circRHOBTB3 (circ_0007444)	Down	qRT-PCR: EOC (*n* = 40), normal ovarian tissues (*n* = 20)	Tissues	FIGO stage, metastasis	OS	-	[110]
		qRT-PCR: OC vs. adjacent normal tissues (*n* = 87)		FIGO stage, grade, tumor size	OS		[111]
circRNA1656 (circ_0002755)	Down	RNA-seq: HGSC (*n* = 3), benign ovarian diseases (*n* = 3). qRT-PCR: HGSC (*n* = 60), ovarian benign diseases (*n* = 60)	Tissues	FIGO stage	-	-	[112]
circSMARCA5 (circ_0001445)	Down	qRT-PCR: OC (36 serous, 14 others), normal ovarian tissues	Tissues	FIGO stage, metastasis	OS	-	[113]
circFBXO7 (circ_0001222)	Down	RNA-seq: EOC (*n* = 27), normal tissues(*n* = 26) qRT-PCR: OC vs. normal tissues (*n* = 12) BaseScope assay: EOC (*n* = 77)	Tissues	-	OS, RFS	-	[114]
circPLEKHM3 (circ_0001095)	Down	RNA-seq: OC vs. normal ovarian tissues (*n* = 5) qRT-PCR: OC vs. normal ovarian tissues (*n* = 12); primary OC vs. matched peritoneal metastatic OC (*n* = 26) BaseScope assay: OC (*n* = 86)	Tissues	Metastasis	OS, RFS	-	[115]
circBNC2 (circ_0008732)	Down	qRT-PCR: EOC vs. benign ovarian cysts vs. healthy (*n* = 83)	Plasma	Tumor grade, subtype, metastasis	-	EOC vs. benign: AUC = 0.879 EOC vs. healthy: AUC = 0.923 Early stage EOC vs. benign: AUC = 0.864 Early stage EOC vs. healthy: AUC = 0.908	[36]
		qRT-PCR: OC vs. adjacent tissues (*n* = 40)	Tissues	FIGO stage, metastasis	OS	-	[116]
circN4BP2L2 *	Down	qRT-PCR: Stage I + II EOC (*n* = 36), stage III + IV EOC (*n* = 90), benign ovarian cysts (*n* = 126), healthy (*n* = 126)	Plasma	FIGO stage, tumor grade, metastasis	-	EOC vs. benign: AUC = 0.82 EOC vs. normal: AUC = 0.90 Early-stage EOC vs. benign: AUC = 0.81 Early stage EOC vs. normal: AUC = 0.90	[43]
circN4BP2L2 (circ_0000471)	Down	RNA-seq: EOC vs. normal ovarian tissues (*n* = 4) qRT-PCR: EOC vs. normal ovarian tissues (*n* = 54)	Tissues	FIGO stage, metastasis	PFS	-	[106]
circFAM120A (circ_0003972) circTOM1L1 (circ_0007288)	Down	Microarray: OC (*n* = 4), vs. uterine myoma patients (*n* = 4) qRT-PCR: OC (*n* = 60: 48 EOC) vs. benign tumors (*n* = 60)	Plasma	Metastasis (circ_0007288)	-	circ_0003972: AUC = 0.724 circ_0007288: AUC = 0.790	[117]

Abbreviations: qRT-PCR, real-time quantitative reverse transcription PCR; OS, overall survival; DFS, disease-free survival; RFS, relapse-free survival; PFS, progression-free survival; ISH, in situ hybridization; FISH, fluorescence in situ hybridization; TNM, tumor node metastasis; DDP, cisplatin; PTX, paclitaxel; SOC, serous ovarian cancer; AUC, area under curve; * circBase ID was not reported by the study; # The circRNA name reported was converted to circBase ID through the website: http://www.bio-inf.cn/CircIDTrans.aspx (accessed on 27 October 2022).

**Table 2 cancers-14-05711-t002:** Effects of circRNAs in EOC.

circRNA (circBase ID)	Experimental Models	Functions	Molecular Mechanisms	Refs
circC20orf11 (circ_0061140)	SKOV3/PTX, HEYA8/PTX Female BALB/c mice s.c. injected with SKOV3/PTX	In vitro: ↑ proliferation/migration/invasion/PTX resistance; ↓ apoptosis In vivo: ↑ tumor growth/PTX resistance	miR-136/CBX2	[125]
	SKOV3, A2780 Male BALB/c mice s.c. injected with SKOV3	In vitro: ↑ proliferation/migration/EMT In vivo: ↑ tumor growth	miR-370/FOXM1	[126]
	SKOV3, A2780 BALB/c mice s.c. injected with SKOV3	In vitro: ↑ proliferation/migration/invasion; ↑ angiogenesis; ↓ apoptosis In vivo: ↑ tumor growth	miR-761/LETM1	[95]
	SKOV3, CAOV3 BALB/c female mice s.c. injected with SKOV3	In vitro: ↑ proliferation/migration/invasion/EMT; ↑ angiogenesis; ↓ apoptosis In vivo: ↑ tumor growth	miR-361-5p/RAB1A	[127]
circC20orf11 *	SKOV3, A2780, SKOV3/DDP, A2780/DDP Male BALB/c mice s.c. injected with A2780/DDP and SKOV3/DDP	In vitro: ↑ proliferation/DDP resistance; ↓ apoptosis In vivo: ↑ tumor DDP resistance	miR-527/YWHAZ	[96]
circCELSR1 *	SKOV3/PTX, A2780/PTX BALB/c mice inoculated with SKOV3/PTX	In vitro: ↑ PTX resistance/viability/proliferation; ↓ apoptosis In vivo: ↑ tumor PTX resistance	miR-149-5p/SIK2	[128]
circCELSR1 (circ_0063809)	SKOV3, HEYA8, SKOV3/PTX, HEYA8/PTX Female BALB/c mice s.c. injected with SKOV3/PTX	In vitro: ↑ PTX resistance In vivo: ↑ tumor PTX resistance	miR-1252/FOXR2	[129]
	SKOV3, A2780 Athymic female nude mice i.p. injected with SKOV3	In vitro: ↑ proliferation/migration/invasion/EMT; ↓ apoptosis In vivo: ↑ tumor growth/metastasis	miR-598/BRD4	[130]
circCELSR1 (circ_0063804)	OVCAR3, SKOV3 Nude mice s.c. injected with OVCAR3	In vitro: ↑ proliferation/DDP resistance; ↓ apoptosis. In vivo: ↑ tumor growth/DDP resistance	miR-1276/CLU	[79]
circFOXM1 (circ_0025033)	SKOV3/PTX, HEYA8/PTX	In vitro: ↑ PTX resistance/migration/invasion; ↓ apoptosis	miR-532-3p/FOXM1	[131]
	SKOV3, A2780 Female BALB/c mice s.c. injected with SKOV3	In vitro: ↑ proliferation/migration/invasion/glycolysis. In vivo: ↑ tumor growth	miR-184/LSM4	[132]
circSEC61A1 (circ_0007841)	A2780/DDP, SKOV3/DDP Female BALB/c mice s.c. injected with SKOV3/DDP	In vitro: ↑viability/proliferation/invasion/migration/DDP resistance; ↓ apoptosis. In vivo: ↑ tumor growth/DDP resistance	miR-532-5p/NFIB	[133]
	SKOV3, OVCAR3 BALB/c mice s.c. injected with SKOV3	In vitro: ↑ proliferation/migration/invasion In vivo: ↑ tumor growth	miR-151-3p/MEX3C	[134]
circMUC16 (circ_0049116)	SKOV3, A2780 BALB/c mice i.p. injected with SKOV3	In vitro: ↑ autophagy/proliferation/invasion In vivo: ↑ tumor metastasis	miR-199a-5p/Beclin1 & RUNX1; ATG13 protein	[84]
	A2780, SKOV3 Female BALB/c mice s.c. injected with A2780	In vitro: ↓ anti-tumor effects of Propofol In vivo: ↑ tumor growth, ↓ anti-tumor effects of Propofol	miR-1182/S100B	[135]
circABCB10 *	SKOV3, UWB1.289	In vitro: ↑ proliferation/invasion; ↓ apoptosis	miR-1271/Wnt/β-catenin	[136]
		In vitro: ↑ proliferation; ↓ apoptosis.	miR-1271, miR-1252, miR-203	[88]
circUBAP2 *	OVCAR3, HO8910	In vitro: ↑ proliferation/migration	miRNA-144/CHD2	[137]
circUBAP2 *	OVCAR3, ES-2	In vitro: ↑ proliferation; ↓ apoptosis.	miR-382-5p/PRPF8	[138]
circPVT1 *	SKOV3, A2780	In vitro: ↑ proliferation/migration/invasion	miR-149-5p/FOXM1	[139]
circPVT1 *	CAOV3, SKOV3	In vitro: ↑ proliferation; ↓ apoptosis.	miR-149	[140]
circRGNEF (circ_0072995)	A2780, HO8910 BALB/c mice s.c. injected with HO8910	In vitro: ↑ proliferation/migration; ↓ apoptosis. In vivo: ↑ tumor growth	miR-147a/CDK6	[141]
	OVCAR3, SKOV3 Female BALB/c mice s.c. injected with OVCAR3	In vitro: ↑ proliferation/migration/invasion; ↓ apoptosis. In vivo: ↑ tumor growth	miR-122-5p/SLC1A5	[142]
circACP6 (circ_0013958)	SKOV3, CAOV3 Female BALB/c mice s.c. injected with SKOV3	In vitro: ↑ proliferation/migration/invasion; ↓ apoptosis. In vivo: ↑ tumor growth	miR-637/PLXNB2	[143]
	A2780, OVCAR3	In vitro: ↑ proliferation/migration/invasion/EMT; ↓ apoptosis		[87]
circNRIP1 (circ_0002711)	A2780/PTX, SKOV3/PTX BALB/c mice s.c. injected with SKOV3/PTX	In vitro: ↑ proliferation/migration/invasion/PTX resistance; ↓ apoptosis. In vivo: ↑ tumor PTX resistance	miR-211-5p/HOXC8	[144]
	SKOV3, OV90 Nude mice s.c. injected with SKOV3	In vitro: ↑ viability/proliferation/glycolysis In vivo: ↑ tumor growth	miR-1244/ROCK1	[145]
circAARS (circ_0000714)	A2780/PTX	In vitro: ↑ PTX resistance/proliferation/cell cycle progression	miR-370-3p/RAB17	[146]
circATL2 (circ_0000993)	HEYA8/PTX, SKOV3/PTX BALB/c mice s.c. infected with SKOV3/PTX	In vitro: ↑ PTX resistance/proliferation; ↓ cell cycle arrest/apoptosis. In vivo: ↑ tumor PTX resistance	miR-506-3p/NFIB	[147]
circTNPO3 (circ_0001741)	SKOV3, HEYA8, SKOV3/PTX, HEYA8/PTX Nude mice s.c. injected with SKOV3/PTX	In vitro: ↑ PTX resistance/cell cycle progression; ↓ apoptosis. In vivo: ↑ tumor PTX resistance	miR-1299/NEK2	[89]
circLPAR3 (circ_0004390)	SKOV3/DDP, A2780/DDP BALB/c mice s.c. injected with SKOV3/DDP	In vitro: ↑ DDP resistance resistances/proliferation/migration/invasion; ↓ apoptosis. In vivo: ↑ tumor growth/DDP resistance	miR-634/PDK1	[148]
	SKOV3, HEYA8, OVCAR429	In vitro: ↑ proliferation	miR-198/MET	[149]
circHIPK2 (circ_0001756)	SKOV3/DDP, A2780/DDP Female BALB/c mice s.c. injected with A2780/DDP	In vitro: ↑ DDP resistance/proliferation/cell cycle progression/migration/invasion; ↓ apoptosis In vivo: ↑ tumor DDP resistance/growth	miR-338-3p/CHTOP	[81]
	SKOV3, A2780 Female BALB/c mice i.v. injected with SKOV3	In vitro: ↑ proliferation/invasion/EMT In vivo: ↑ tumor growth	IGF2BP2/EGFR/MAPK	[150]
circFOXP1 (circ_0001320)	COC1, SKOV3, SKOV3/DDP Nude mice s.c. injected with SKOV3/DDP	In vitro: ↑ proliferation/DDP resistance In vivo: ↑ tumor growth/DDP resistance	miR-22 & miR-150-3p/CEBPG & FMNL3	[99]
circPRKCI (circ_0067934)	A2780/DDP Female BALB/c mice s.c. injected with A2780/DDP	In vitro: ↑ DDP resistance/proliferation/invasion; ↓ apoptosis In vivo: ↑ tumor growth	miR-545-3p/PPA1/JNK	[151]
circPBX3 (circ_0004804)	OV90, SKOV3; OV90-Res; SKOV3-Res Female nude mice s.c. injected with OV90	In vitro: ↑ DDP resistance In vivo: ↑ tumor DDP resistance	IGF2BP2/ATP7A	[152]
circZFR*	A2780	In vitro: ↑ proliferation/migration/invasion/glycolysis; ↓ apoptosis	miR-212-5p/SOD2	[153]
circE2F2 (circ_0000030)	OVCAR3, SKOV3 Female BALB/c mice s.c. injected with OVCAR3	In vitro: ↑ proliferation/migration/invasion/ glucose. In vivo: ↑ tumor growth	HuR/E2F2.	[154]
circPTK2 (circ_0008305)	SKOV3, OVCAR3 Female nude mice i.p. injected with SKOV3	In vitro: ↑ migration/invasion/EMT; ↑ angiogenesis In vivo: ↑ tumor growth	miR-639/FOXC1	[155]
circASH2L (circ_0006302)	A2780, SKOV3 Female BALB/c mice s.c. injected with SKOV3	In vitro: ↑ invasion/proliferation; ↑angiogenesis In vivo: ↑ tumor angiogenesis/lymphangiogenesis/growth	miR-665/VEGFA	[86]
circITGB6 (circ_0056856)	OVCAR3, CAOV3, ID8 (mice) Female C57BL/6 mice i.p. injected with ID8	In vitro: ↑ DDP resistance In vivo: ↑ M2 macrophage-dependent DDP resistance of tumor	IGF2BP2/FGF9	[97]
circMGAT5 (circ_0001068)	A2780	In vitro: ↑ PD1 expression in T cells, ↑ T cell exhaustion	miR-28-5p/PD1	[100]
circSPECC1 (circ_0000745)	ES-2, SKOV3	In vitro: ↑ proliferation/migration/invasion/EMT/stemness	miR-3187-3p/ERBB4/PI3K/AKT	[80]
circTUBA1B (circ_0026123)	SKOV3 Female BALB/c mice s.c. injected with SKOV3	In vitro: ↑ migration/proliferation/cancer stem cell differentiation In vivo: ↑ tumor growth	miR-124-3p/EZH2	[156]
circRAB11FIP1 (circ_0005630)	SKOV3, A2780 BALB/c mice i.p. injected with SKOV3	In vitro: ↑ autophagic flux/proliferation/invasion In vivo: ↑ tumor metastasis	miR-129/ATG14 & ATG7; DSC1/ATG101	[157]
circFGFR3 *	SKOV3, A2780 Mice inoculated with SKOV3	In vitro: ↑ proliferation/migration/invasion/EMT In vivo: ↑ tumor growth/metastasis	miR-29a-3p/E2F1	[158]
circCSPP1 (circ_0001806)	OVCAR3, A2780, CAOV3	In vitro: ↑ proliferation/migration/invasion/EMT	miR-1236-3p/ZEB1	[75]
circKRT7 (circ_0026360)	ES-2, SKOV3 BALB/c mice s.c. injected with ES-2	In vitro: ↑ proliferation/migration/invasion/EMT In vivo: ↑ tumor growth	miR-29a-3p/COL1A1	[159]
circLIN52 (circ_0000554)	HO8910	In vitro: ↑ proliferation/invasion/EMT	miR-567	[160]
circSLAIN1 (circ_0000497) circGMIP (circ_0000918)	SKOV3, OVCAR3	In vitro: ↑ migration/invasion/EMT	predicted miRNAs/mRNAs	[161]
circEPSTI1 (circ_0000479)	OV119, A2780 BALB/c mice injected (s.c. and i.v.) with OV119	In vitro: ↑ proliferation/invasion; ↓ apoptosis In vivo: ↑ tumor growth/metastasis	miR-942/EPSTI1	[162]
circGFRA1 *	OV119, A2780 BALB/c mice injected (s.c. and i.v.) with OV119	In vitro: ↑ proliferation/invasion In vivo: ↑ tumor growth/metastasis	miR-449a/GFRA1	[163]
circMFN2 (circ_0009910)	SKOV3	In vitro: ↑ proliferation/migration/invasion	miR-145/NF-κB & Notch	[82]
circCEACAM5 (circ_0051240)	OVCAR3, HO8910 Male nude mice, s.c. injected with OVCAR3	In vitro: ↑ proliferation/migration/invasion In vivo: ↑ tumorigenesis	miR-637/KLK4	[74]
circPIP5K1A (circ_0014130)	SKOV3, A2780 BALB/c mice s.c. injected with SKOV3	In vitro: ↑ proliferation/migration/invasion In vivo: ↑ tumor growth	miR-661/IGFBP5	[164]
circRNA_102958 (circ_0003854) #	SKOV3, OVCAR3	In vitro: ↑ proliferation/migration/invasion	miR-1205/SH2D3A	[165]
circFAM53B (circ_0000267)	A2780, HO8910	In vitro: ↑ proliferation/migration/invasion; ↓ apoptosis	miRNA-646/VAMP2, miRNA-647/MDM2	[85]
circSETDB1 (circ_0006352)	A2780, SKOV3 Female BALB/c mice i.p. injected with SKOV3	In vitro: ↑ proliferation/invasion/migration; ↓ apoptosis In vivo: ↑ tumor growth	miR-129-3p/MAP3K3	[166]
circPDE7B (circ_0004712)	OVCAR3, SKOV3 Female BALB/c mice s.c. injected with SKOV3	In vitro: ↑ proliferation/invasion/migration; ↓ apoptosis In vivo: ↑ tumor growth	miR-331-3p/FZD4	[94]
circMYLK *	A2780, CAOV3	In vitro: ↑ proliferation	miR-652	[83]
circPGAM1 (circ_0019340)	CAOV3, OVCAR3 Nude mice s.c. injected with CAOV3 and OVCAR3	In vitro: ↑ proliferation/migration/invasion; ↓ apoptosis In vivo: ↑ tumor growth	miR-542-3p/CDC5L/PEAK1	[167]
circFOXO3 *	SKOV3	In vitro: ↑ proliferation/migration/invasion	miR-422a/PLP2	[168]
circRNA_051239 (circ_0051239) #	SKOV3, SKOV3.ip	In vitro: ↑ proliferation/migration/invasion	miR-509-5p/PRSS3	[169]
circPUM1 (circ_0000043)	A2780, CAOV3 Female BALB/c mice i.p. injected with A2780 and CAOV3	In vitro: ↑ proliferation/migration/invasion; ↓ apoptosis; ↑ MMT of peritoneal mesothelial cells In vivo: ↑ tumor growth/metastasis	miR-615-5p/NF-κB, miR-6753-5p/MMP2	[77]
circWHSC1 (circ_0001387)	CAOV3, OVCAR3 Female BALB/c mice injected (s.c. or i.p.) with CAOV3.	In vitro: ↑ proliferation/migration/invasion; ↓ apoptosis; ↑ MMT of peritoneal mesothelial cells In vivo: ↑ tumor growth/metastasis	miR-145 & miR-1182/MUC1 & hTERT	[170]
circRHOC (circ_0013549)	A2780, CAOV3 Female BALB/c mice i.p. injected with A2780	In vitro: ↑ viability/migration/invasion In vivo: ↑ tumor metastasis	miR-302e/VEGFA; VEGFA	[78]
circNOLC1 (circ_0000257)	CAOV3, A2780 Female BALB/c mice s.c. injected with A2780	In vitro: ↑ proliferation/migration/invasion; ↓ apoptosis In vivo: ↑ tumor growth	ESRP1/CDK1/RhoA	[76]
circCRIM1 (circ_0002346)	OVCAR3, CAOV3 Female BALB/c mice s.c. injected with CAOV3	In vitro: ↑ viability/migration/invasion; ↓ apoptosis In vivo: ↑ tumor growth	miR-145-5p/CRIM1, miR-383-5p/ZEB2; encoding a 188aa protein	[171]
circVPS13C*	A2780, SKOV3	In vitro: ↑ proliferation/migration/invasion/cell cycle progression; ↓ apoptosis	miR-145/MEK/ERK	[172]
circSLAMF6 (circ_0000144)	SKOV3, ES-2 Female BALB/c mice s.c. injected with SKOV3	In vitro: ↑ proliferation/invasion/migration In vivo: ↑ tumor growth	miR-610/ELK3	[93]
circRNF144B (circ_0075797)	SKOV3, OVCAR3 Mice s.c. injected (s.c. and i.v.) with SKOV3	In vitro: ↓ autophagy; ↑ proliferation/migration/invasion In vivo: ↑ tumor growth/metastasis	miR-342-3p/FBXL11/Beclin-1	[92]
circANKRD17 (circ_0007883)	A2780, SKOV3, A2780/PTX, SKOV3/PTX BALB/c female mice s.c. injected with SKOV3/PTX	In vitro: ↑ PTX chemoresistance In vivo: ↑ tumor growth/PTX chemoresistance	FUS/FOXR2	[90]
	A2780, SKOV3	In vitro: ↑ proliferation/invasion/migration/EMT; ↓ apoptosis	ZEB1/circANKRD17	[173]
circEEF2 (circ_0048559)	SKOV3, A2780 BALB/c mice s.c. injected with SKOV3, A2780 and i.p. injected with SKOV3	In vitro: ↑ autophagy/proliferation/invasion In vivo: ↑ tumor growth/metastasis	miR-6881-3p/ATG5 & ATG7, ANXA2/p-mTOR	[91]
circCFH (circ_0015756)	OV90, SKOV3 Female BALB/c mice s.c. injected with SKOV3	In vitro: ↑ proliferation/migration/invasion; ↓ apoptosis In vivo: ↑ tumor growth	miR-942-5p/CUL4B	[174]
circKIF4A (circ_0007255)	CAOV3, SKOV3 Female BALB/c mice injected (s.c. and i.v.) with EOC cells	In vitro: ↑ proliferation/invasion In vivo: ↑ tumor growth/metastasis	miR-127/JAM3	[175]
circEXOC6B (circ_0009043)	A2780, SKOV3 BALB/c mice s.c. injected with SKOV3	In vitro: ↓ proliferation/migration/invasion/PTX resistance In vivo: ↓ tumor PTX resistance	miR-376c-3p/FOXO3	[105]
circEXOC6B*	A2870, SKOV3	In vitro: ↓ proliferation/invasion; ↑ apoptosis	miR-421/RUS1	[176]
circPLEKHM3 (circ_0001095)	A2780, MDAH2274, OV90 Female BALB/c mice s.c. injected with A2780	In vitro: ↓ proliferation/migration/EMT In vivo: ↓ tumor growth	miR-9/BRCA1/DNAJB6/KLF4/AKT	[115]
	SKOV3, A2780 Female BALB/c mice s.c. injected with A2780	In vitro: ↓ proliferation; ↑ apoptosis; ↑ anti-tumor effect of curcumin In vivo: ↓ tumor growth; ↑ anti-tumor effect of curcumin	miR-320a/SMG1	[177]
circCDR1as (ciRS-7, circ_0001946)	SKOV3, A2780 Male BALB/c mice s.c. injected with SKOV3	In vitro: ↑ proliferation/migration/invasion In vivo: ↑ tumor growth	miR-641/ZEB1 & MDM2	[69]
	SKOV3, SKOV3/CDDP, HO8910, HO8910/CDDP Female BALB/c mice s.c. injected with SKOV3	In vitro: ↓ DDP resistance resistance/proliferation/migration/invasion; ↑ apoptosis In vivo: ↓ tumor growth/DDP resistance	miR-1299/PPP1R12B; AKT/mTOR	[70]
	A2780, SKOV3, A2780-DDP, SKOV3-DDP BALB/c female mice s.c. injected with SKOV3	In vitro: ↓ DDP resistance/proliferation/migration; ↑ apoptosis In vivo: ↓ tumor growth/DDP resistance	miR-1270/SCAI	[71]
	A2780, HO8910	In vitro: ↓ proliferation/invasion/migration	miR-135b-5p/HIF1AN	[72]
circRHOBTB3 *	A2780, OV90 Female BALB/mice s.c. injected with A2780	In vitro: ↓ proliferation/migration/invasion/glycolysis/EMT In vivo: ↓ tumor growth/metastasis	PI3K/AKT	[178]
circRHOBTB3 (circ_0007444)	SKOV3, OVCAR8 BALB/c mice s.c. injected with SKOV3	In vitro: ↓ proliferation/G1/S transition/invasion; ↑ apoptosis In vivo: ↓ tumor growth	miR-23a-3p/PTEN/AKT	[110]
	SKOV3, OVCAR3 Female nude mice s.c. injected with SKOV3, OVCAR3	In vitro: ↓ proliferation/migration/invasion; ↑ apoptosis. In vivo: ↓ tumor growth	miR-570-3p/PTEN	[111]
circATRNL1 (circ_0020093)	SKOV3, CAOV3 Nude mice i.p. injected with SKOV3 and CAOV3	In vitro: ↓ proliferation/migration/invasion; ↓ angiogenesis; ↑ apoptosis In vivo: ↓ tumor growth/metastasis	miR-378/SMAD4	[107]
	A2780, SKOV3, SW626	In vitro: ↓ proliferation/migration/invasion	encoding a 131 aa protein	[179]
circITCH *	SKOV3, CAOV3 Female BALB/c mice s.c. injected with SKOV3	In vitro: ↓ proliferation/migration/invasion In vivo: ↓ tumor growth	miR-145/RASA1	[180]
circITCH *	SKOV3	In vitro: ↓ proliferation; ↑ apoptosis	miR-10a	[181]
circITCH *	UWB1.289 + BRCA1, UWB1.289	In vitro: ↓ proliferation	lncRNA HULC	[182]
circITCH *	A2780, OVCAR3 BALB/c mice s.c. injected with OVCAR3	In vitro: ↓ proliferation/invasion/glycolysis; ↑ apoptosis In vivo: ↓ tumor growth	miR-106a/CDH1	[101]
circSLC22A3 (circ_0078607)	HEY, ES-2 Female BALB/c mice s.c. injected with HEY	In vitro: ↓ proliferation/migration/invasion; ↑ apoptosis In vivo: ↓ tumor growth	miR-32-5p/SIK1	[108]
	SKOV3, A2780	In vitro: ↓ proliferation; ↑ apoptosis	miR-518a-5p/Fas	[183]
circRNA_100395 (circ_0015278) #	SKOV3, ES-2	In vitro: ↓ proliferation/migration/invasion/EMT	miR-1228/p53	[103]
circLARP4 *	SKOV3, A2780	In vitro: ↓ proliferation/migration/invasion; ↑ apoptosis	miR-513b-5p/LARP4	[184]
circFBXO7 (circ_0001222)	A2780, MDAH2774, SKOV3, OV90 Female BALB/c mice s.c. injected with MDAH2774, NOD/SCID mice i.p. injected with MDAH2774	In vitro: ↓ proliferation/migration/invasion In vivo: ↓ tumor growth/metastasis	miR-96-5p/MTSS1/Wnt/β-catenin	[114]
circBNC2 (circ_0008732)	SKOV3, HO8910	In vitro: ↓ proliferation/migration/invasion	miR-223-3p/FBXW7	[116]
circ_9119 *	ES-2, SKOV3 Female BALB/c mice s.c. injected with SKOV3	In vitro: ↓ proliferation; ↑ apoptosis In vivo: ↓ tumor growth	miR-21/PTEN/AKT	[185]
circZNF608 (circ_0001523)	A2780, SKOV3, SW626	In vitro: ↓ proliferation/migration/invasion	miR-152-5p	[179]
circMTO1 *	SKOV3, OVCAR3	In vitro: ↓ proliferation/invasion	miR-182-5p/KLF15	[186]
circMTO1 (circ_0007874)	A2780, SKOV3 BALB/c mice s.c. injected with A2780	In vitro: ↓ migration/proliferation In vivo: ↓ tumor growth	miR-760/SOCS3	[187]
circHIPK3 (circ_0000284)	A2780, SKOV3	In vitro: ↓ proliferation/migration/invasion; ↑ apoptosis	predicted miRNAs/mRNAs	[188]
circSMARCA5 (circ_0001445)	SKOV3, HO8910 Female BALB/c mice injected (s.c. and i.p.) with HO8910	In vitro: ↓ proliferation/invasion/migration; ↑ apoptosis In vivo: ↓ tumor growth/metastasis	miR-576-5p/SFRP1/WNT/β-catenin	[113]
circCERS6 (circ_0007024)	SKOV3, OVCAR3 Nude mice s.c. injected with SKOV3	In vitro: ↓ proliferation/invasion/migration/EMT In vivo: ↓ tumor growth	miR-630/RASSF8	[189]

Abbreviations: EMT, epithelial-mesenchymal transition; MMT, mesothelial-to-mesenchymal transition; DDP, cisplatin; PTX, paclitaxel; s.c., subcutaneous injection; i.p., intraperitoneal injection; i.v., intravenous injection; ↑ increased; ↓ decreased; * circBase ID was not reported by the study; # The circRNA name reported was converted to circBase ID through the website: http://www.bio-inf.cn/CircIDTrans.aspx (accessed on 27 October 2022).

## Data Availability

Not applicable.

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
