# Peer review of "Circular RNAs in Epithelial Ovarian Cancer: From Biomarkers to Therapeutic Targets"

_cancers, 2022, doi:10.3390/cancers14225711_

Round 1

Reviewer 1 Report

The authors briefly described the potential significance of dysregulated circRNAs as biomarkers and therapeutic targets in EOC. Especially when circRNAs are mentioned as potential therapeutic targets, this review detailed the promoting and supressing effects exerted by circRNAs in EOC, which was from an circRNAs perspective. However, as the authors stated in the manuscript, there have been a large number of reviews which focus on the biological role of aberrantly expressed circRNAs played in malignancies.  Herein, one suggestion I have is that the authors could elaborate respectively on the impact of circRNAs from a clinical perspective, such as chemoresistance, lymph node metastasis, ascites formation, etc. For instance, chemoresistance,  especially to platinum-based chemotherapy, is a key cause of EOC relapse and poor prognosis, and as the previous study showed, circRNAs could confer to its resistance via various mechanisms, like sponges, RBPs, peptides coding, etc. So that it might be more relevant and novel.

Here are a few other small questions:

1.Page2, Line19:"EOC can be classified into high-grade serous, low-grade serous, endometroid, mucinous, and clear cell carcinoma"-Regarding to WHO classfication, the epithelial ovarian tumors should also include undifferentiated carcinomas, mixed epithelial tumors and Brenner tumors. Whether these pathological subtypes of tumors were included in the review?

2.Currently, ASOs, the modified antisense oligonucleotides which could specifically target the juction site of circRNA, demonstrated good anti-tumor effects in vivo. What is the recent status of research on this therapy in ovarian cancer? What are its prospects as an circRNA target? Please add to the manuscript as appropriate.

Author Response

Thank you very much for your thoughtful comments and suggestions.  We have made revisions according to your suggestions.

Comment:

The authors briefly described the potential significance of dysregulated circRNAs as biomarkers and therapeutic targets in EOC. Especially when circRNAs are mentioned as potential therapeutic targets, this review detailed the promoting and suppressing effects exerted by circRNAs in EOC, which was from a circRNAs perspective. However, as the authors stated in the manuscript, there have been a large number of reviews which focus on the biological role of aberrantly expressed circRNAs played in malignancies.  Herein, one suggestion I have is that the authors could elaborate respectively on the impact of circRNAs from a clinical perspective, such as chemoresistance, lymph node metastasis, ascites formation, etc. For instance, chemoresistance,  especially to platinum-based chemotherapy, is a key cause of EOC relapse and poor prognosis, and as the previous study showed, circRNAs could confer to its resistance via various mechanisms, like sponges, RBPs, peptides coding, etc. So that it might be more relevant and novel.

Response: We focus our manuscript mainly on the clinical relevance of circRNAs for EOC diagnosis/ prognosis and their therapeutic potential. We expanded the discussion, wherever possible in the revised manuscript.

Here are a few other small questions:

  1. Page 2, Line 19: "EOC can be classified into high-grade serous, low-grade serous, endometroid, mucinous, and clear cell carcinoma"-Regarding WHO classification, the epithelial ovarian tumors should also include undifferentiated carcinomas, mixed epithelial tumors and Brenner tumors. Whether these pathological subtypes of tumors were included in the review?

Response: We added a sentence to state the existence of these rare subtypes but did not discuss them in the review as there are no related studies.  

  1. Currently, ASOs, the modified antisense oligonucleotides which could specifically target the junction site of circRNA, demonstrated good anti-tumor effects in vivo. What is the recent status of research on this therapy in ovarian cancer? What are its prospects as a circRNA target? Please add to the manuscript as appropriate.

Response: So far there is one study in EOC that has used ASO to knock down circRNA. We have added a few sentences in the “Concluding remarks and future directions” to discuss this.

Reviewer 2 Report

the manuscript is very interesting, well illustrated and generally well written. To my opinion, it can be accepted in the present form.

Author Response

Thank you

Reviewer 3 Report

It was a nice paper about the application of Circular RNAs for the treatment and diagnosis of Epithelial ovarian cancer. Here are just two comments to improve the quality of your paper:

1-      Please use “hyphen (-)” instead of “dash (/)” in table 1.

2-      Please introduce all the abbreviations in their first-time usage.

Author Response

Thank you for your comments and suggestions.  

Dashes in Table 1 have been replaced by hyphens. All abbreviations, except for gene/protein symbols, are now defined.